# Postmortem Minimally Invasive Autopsy in Critically Ill COVID-19 Patients at the Bedside: A Proof-of-Concept Study at the ICU

**DOI:** 10.3390/diagnostics14030294

**Published:** 2024-01-30

**Authors:** Tobias Lahmer, Gregor Weirich, Stefan Porubsky, Sebastian Rasch, Florian A. Kammerstetter, Christian Schustetter, Peter Schüffler, Johanna Erber, Miriam Dibos, Claire Delbridge, Peer Hendrik Kuhn, Samuel Jeske, Manuel Steinhardt, Adam Chaker, Markus Heim, Uwe Heemann, Roland M. Schmid, Wilko Weichert, Konrad Friedrich Stock, Julia Slotta-Huspenina

**Affiliations:** 1Department of Internal Medicine II, Klinikum Rechts der Isar, School of Medicine, Technical University of Munich, Ismaninger Straße 22, 81675 Munich, Germany; sebastian.rasch@mri.tum.de (S.R.); johanna.erber@mri.tum.de (J.E.); miriam.dibos@mri.tum.de (M.D.); rolandm.schmid@mri.tum.de (R.M.S.); 2Institute of Pathology, School of Medicine, Technical University Munich, Ismaninger Straße 22, 81675 Munich, Germany; gregor.weirich@tum.de (G.W.); florian.kammerstetter@epar.at (F.A.K.); christian.schustetter@tum.de (C.S.); peter.schueffler@tum.de (P.S.); c.delbridge@tum.de (C.D.); peer-henrik.kuhn@tum.de (P.H.K.); wilko.weichert@mri.tum.de (W.W.); julia.slotta-huspenina@tum.de (J.S.-H.); 3Institut für Pathologie, Universitätsklinikum Mainz, Langenbeckstraße 1, 55131 Mainz, Germany; stefan.porubsky@unimedizin-mainz.de; 4Institute of Virology, School of Medicine, Technical University of Munich/Helmholtz Zentrum München, Trogerstraße 30, 81675 Munich, Germany; samuel.jeske@tum.de; 5Department of Diagnostic and Interventional Radiology, School of Medicine, Technical University of Munich, Ismaninger Straße 22, 81675 Munich, Germany; manuel.steinhardt@tum.de; 6Department of Otorhinolaryngology, University Hospital Klinikum Rechts der Isar, Ismaninger Straße 22, 81675 Munich, Germany; adam.chaker@mri.tum.de; 7Department of Anesthesiology and Intensive Medicine, School of Medicine, Technical University Munich, Ismaninger Straße 22, 81675 Munich, Germany; markus.heim@mri.tum.de; 8Department of Nephrology, School of Medicine, Technical University Munich, Ismaninger Straße 22, 81675 Munich, Germany; uwe.heemann@mri.tum.de (U.H.); konrad.stock@mri.tum.de (K.F.S.)

**Keywords:** minimal invasive autopsy, COVID-19, ICU, postmortem diagnostic

## Abstract

Background: Economic restrictions and workforce cuts have continually challenged conventional autopsies. Recently, the COVID-19 pandemic has added tissue quality and safety requirements to the investigation of this disease, thereby launching efforts to upgrade autopsy strategies. Methods: In this proof-of-concept study, we performed bedside ultrasound-guided minimally invasive autopsy (US-MIA) in the ICU of critically ill COVID-19 patients using a structured protocol to obtain non-autolyzed tissue. Biopsies were assessed for their quality (vitality) and length of biopsy (mm) and for diagnosis. The efficiency of the procedure was monitored in five cases by recording the time of each step and safety issues by swabbing personal protective equipment and devices for viral contamination. Findings: Ultrasound examination and tissue procurement required a mean time period of 13 min and 54 min, respectively. A total of 318 multiorgan biopsies were obtained from five patients. Quality and vitality standards were fulfilled, which not only allowed for specific histopathological diagnosis but also the reliable detection of SARS-CoV-2 virions in unexpected organs using electronic microscopy and RNA-expressing techniques. Interpretation: Bedside multidisciplinary US-MIA allows for the fast and efficient acquisition of autolytic-free tissue and offers unappreciated potential to overcome the limitations of research in postmortem studies.

## 1. Introduction

Since the beginning of the SARS-CoV-2 pandemic, tremendous worldwide efforts have helped develop treatment options for patients with COVID-19 and understand the nature and possible long-term effects of infection with this new coronavirus [1,2]. The severity of COVID-19 may vary according to different SARS-CoV-2 variants of concern (VOC), ranging from typical pulmonary manifestations to the affection of extrapulmonary sites such as cerebral, myocardial, renal, and gastrointestinal sites. These organ changes may either deteriorate into acute multiorgan failure or lead to a chronic “Post-COVID-19” syndrome [2,3,4,5]. Easy-to-implement genetic and molecular blood tests have helped predict the organ-specific severity of the SARS-CoV-2 infection [6]. In contrast, to track the detailed pathogenesis of COVID-19, multiorgan tissue analysis is still of prime importance.

Conventional autopsies helped at the beginning of the pandemic to define COVID-19-related organ damage, such as thromboembolism and diffuse alveolar damage [7,8,9,10]. Despite these new insights, autopsy rates remained low during the pandemic, leaving our knowledge of COVID-19 disrupted by considerable gaps [11,12].

This may be supported by the typical shortcomings of conventional autopsies, such as the misperception of medical use and lack of universal standard operating procedures (SOPs) [13]. In addition, improvements in imaging modalities, the growing body of biomarkers, and molecular pathological and microbial research have contributed to this development to track the biology of disease states in living organisms, facilitating theranostics [14,15,16].

Alternative procedures for postmortem tissue sampling with a short postmortem interval (PMI) are needed to use the improved methodology of DNA/RNA analyses, next-generation sequencing (NGS), and cell culture experiments, including organoids, to track diseases at the molecular level and adequately reevaluate the concepts of etiology and pathogenesis [17]. In this context, minimally invasive approaches in combination with high-end imaging techniques (especially ultrasound and computed tomography (CT)) are ideal for obtaining postmortem specimens of such quality without spreading the environment with a highly contagious new agent. Accordingly, a minimally invasive tissue sampling (MITS) strategy has been advocated as a safe method.

Not only are tissue samples vital shortly after death (PMI: 1–8 h) in most cases, but postmortem needle biopsies can reach almost any organ and structure under ultrasound-guided visual control, enabling research on remote organs and body sites that are usually not targeted during conventional autopsies (i.e., salivary and lacrimal glands, eye, conjunctiva, and penis) [18,19]. In addition, this approach is faster and cheaper than conventional autopsies when performed by a well-trained and experienced operator.

To highlight these benefits, we performed a proof-of-concept study in a multidisciplinary approach using bedside minimally invasive tissue sampling directly at the intensive care unit (ICU) in critically ill COVID-19 patients in the nearest possible timeframe after their death.

## 2. Material and Methods

### 2.1. Study Design

This proof-of-concept study evaluated the organizational feasibility, efficiency, and safety of bedside postmortem minimally invasive tissue sampling in the intensive care unit of Medizinische Klinik und Poliklinik II, Klinikum Rechts der Isar of the Technical University of Munich (TUM), a tertiary care center in Germany (Figure 1). Patient enrollment for this study started on 10 December 2021, and ended on 15 January 2022. The inclusion criteria were SARS-CoV-2 infection confirmed by reverse transcription polymerase chain reaction (RT-PCR) on nasopharyngeal swabs or bronchoalveolar lavage during the hospital stay and next of kin’s consent. Biopsy samples were assessed for tissue quality and diagnosis. The time span of each investigational step was recorded to monitor the efficiency of the procedure. For safety reasons, viral contamination of personal protective equipment (PPE) and devices was monitored using swabs at the end of the biopsy procedures. This study was approved by the ethics committee of the TUM (Ref. 225/20S; date: 13 December 2021) and was performed in accordance with the ethical principles of the Helsinki Declaration. The basis for conducting a clinical autopsy is the consent of the relatives, whereby there are neither nationwide nor statewide uniform regulations regarding the specific procedure. The consent process at the MRI involves informing and obtaining consent from the nearest relative, which must be documented in writing by the relative’s signature on the corresponding form. Deceased individuals without relatives or deceased individuals with legal guardians without a guardianship directive beyond death cannot, in principle, undergo an autopsy according to the specific local regulations of the München rechts der Isar (MRI).

The consent of the next of kin for an autopsy of a deceased person due to COVID-19 may be given orally (e.g., over the phone) and does not require a personal signature. The written documentation of consent may be carried out by the informing physician, providing the name of the relative, contact information, and the date and time of the consent.

This procedure was confirmed during the COVID-19 pandemic situation by the local ethics committee of the TUM (see above), Munich, Bavaria, Germany.

### 2.2. Logistical Aspects and Standard Operating Procedures

An interdisciplinary team of intensive care physicians, pathologists, and ultrasound specialists defined the standard operating procedures. The minimal invasive autopsy (MIA) was performed by the tissue procurement team (TPT), consisting of a pathologist (tasks: needle biopsy, macroscopic quality assessment, and microscopic pathology), an ultrasound specialist (tasks: ultrasound examination and biopsy guidance), and a technician (tasks: preparation of pseudonymized sampling protocols and biopsy containers and specimen handling), and was on standby for 12 h (from 8 a.m. to 8 p.m.), 7 days a week. The procedure was performed in the rooms of ICU patients at the deceased’s bedsides. After each patient died, their relatives were informed of the procedure. After oral consent was obtained, the TPT was informed, and patient-specific sampling protocols and biopsy containers were prepared, including appropriate pseudonymization. Before the arrival of the TPT in the ICU, the patient’s room was prepared using portable room dividers in multibed rooms. Clinical, imaging, and laboratory data were obtained from the medical charts and hospital information systems. Signs of death were ascertained and documented by a physician who was not involved in this study.

Ultrasound examination and sampling protocol: A Siemens Acuson S 3000 ultrasound system (Siemens Medical Solutions, Mountain View, CA, USA) with a 4–9 MHz linear transducer and 1–4 MHz convex probe was used. The deceased patient remained in the supine position during the procedure, and the arms were placed above the head to expose the lateral thorax for sampling. Ultrasound examinations involve measuring organ size using the leading-edge method and determining pathological changes. Lung ultrasounds were performed according to a standardized protocol for COVID-19 patients [18]. Biopsies were retrieved under full visual control in real time using an in-plane multi-angle needle guidance system (Ultra-Pro II; Civco, Coralville, IA, USA). A single-use biopsy device was used (14 G; 20 cm length; 2.2 cm stroke length; PlusSpeed^®^, Pflugbeil, Zorneding, Germany). The ultrasound specialist localized the target to determine the needle trajectory and indicated the correct position for launching the biopsy gun to the pathologist. Finally, the pathologist evaluated biopsy adequacy by immediate macroscopic inspection to prompt a rebiopsy when inadequate. Adequacy was assumed if the specimen size was satisfactory for an analysis of 5–10 qm. Regularly sampled organs included the lungs (at least three areas per side), liver (at least segments III, IV, and VIII), spleen, pancreas, kidneys, left heart, abdominal aorta, parotid gland, conjunctival mucosa, lacrimal gland, skin, and bone marrow. The conjunctival mucosa and lacrimal gland were obtained from the left lateral oculi angulus using forceps and scissors, and the skin and bone marrow were sampled from the left anterior iliac crest without ultrasound guidance. Further samples of the presumably pathologically altered organs were collected based on abnormal imaging findings or clinical information.

Biopsy processing protocol: Three needle passes were performed from each organ/puncture site with a minimum distance of 1 cm from each other. One sample was immediately fixed with 10% neutral-buffered formalin (at least 72 h for viral inactivation) and paraffin-embedded (FFPE) for diagnostic purposes according to the standard procedures of the institutes of pathology (DIN ISO 17020). Two other samples were collected for biobanking and research purposes and preserved using the PAXgene^®^ tissue system (Qiagen, Hilden, Germany) and cryoconservation (−80 °C) according to the standard procedures of the biobank of the Klinikum rechts der Isar, MTBio (DIN ISO 20387). FFPE biopsies were sectioned and stained with hematoxylin and eosin (H&E), periodic acid-Schiff (PAS), Elastica van Gieson’s stain (EvG), Ladewig, Prussian blue, Gomori, and chloroacetate esterase. Immunostaining was performed for selected cases (CD34, CD68, CMV, HSV, and CK7/p40). All slides were then digitized using a Leica GT450Dx device (Leica Biosystems, Buffalo Grove, IL, USA) at 40× (0.25 µm/px). Some samples were investigated through electron microscopy (EM) and for SARS-CoV-2 genomic RNA through in situ hybridization. For EM tissues, they were post-fixed in 2.5% glutaraldehyde (Serva, Heidelberg, Germany), followed by 1% osmium tetraoxide (Carl Roth, Karlsruhe, Germany), and embedded in Epon resin (Agar Scientific, Stansted, UK). Images of 80 nm thin sections were taken using the transmission EM JEM-1400 (Joel, Freising, Germany). In situ hybridization was performed on FFPE samples using RNAscope^®^ 2.5 HD Detection Reagents-RED (Cat. No. 322350, Advanced Cell Diagnostics, ACD, Bio-Techne, Newark, CA, USA) and SARS-CoV-2 RNAscope^®^ ISH Probe (21,631–23,303 Cat. No. 848561, ACD) according to the manufacturer’s protocol. In brief, deparaffinized sections were subjected to target retrieval for 20 min at 98–102 °C in 1× Target Retrieval Solution and to Protease Plus treatment for 15 min at 40 °C in an HybEZ oven (ACD). A 50% gill hematoxylin I (Sigma-Aldrich, St. Louis, MO, USA) staining solution was used for counterstaining for 2 min at room temperature. Two samples were used as negative and positive probe controls to check the assay’s specificity. Two general pathologists with expertise in autopsy pathology (GW and JSH) analyzed the FFPE biopsies; EM was analyzed by a pathologist with special expertise (SP).

### 2.3. Assessment Tools for Time Efficiency and Sample Quality

To address time efficiency, the following time points were documented: patient death, consent obtained, TPT informed, ultrasound examination initiated, sampling initiated, and tissue collection completed. The time between the patient’s death and the fixation of the biopsy is referred to as the PMI. For quality control, the biopsy length (mm), signs of autolysis (yes or no), and representativity were assessed. Representativeness was assumed if organ-specific tissue of the target organ was captured.

### 2.4. Safety/Hygiene

The personnel involved wore protective equipment, including eye protection equipment, two layers of gloves, plastic arm sleeves, shoe covers, and FFP2 or FFP3 masks. During the procedure, the room was reserved for the MIA. After each biopsy, the biopsy device was decontaminated using Incidin™ Plus (0.5%) immersion for 10–15 s and rinsed in tap water. After the procedure, the personnel’s decontamination and the decontamination of the devices used occurred according to standard hygienic protocols concerning the handling of SARS-CoV-2-wetted materials. To assess SARS-CoV-2 contamination of PPE or the devices used, swabs were taken after the completion of biopsy collection and before decontamination. The (right-handed) sonographer, pathologist, and devices were swabbed thoroughly in a meandering manner for at least 10 s at various sites (see Figure 2 for the swab collection locations). Nasopharyngeal swabs from the deceased were used as the controls. Commercially available swab sets (Clinical Virus Transport Medium; Noble Biosciences) were used. Samples were stored in refrigerators at 4 °C until PCR analysis. Detection of SARS-CoV-2 RNA was performed on a Roche Cobas 6800 analyzer using a Cobas SARS-CoV-2 Test Kit (Roche Diagnostics, Mannheim, Germany). Results were reported using the Cobas system as the threshold cycle (Ct) value for the targeted SARS-CoV-2 ORF1/a and E genes. The viral load was quantified using a standard curve determined according to the World Health Organization’s (WHO’s) International Standard for SARS-CoV-2 RNA (WHO/BS/2020.2402).

## 3. Results

Six COVID-19 patients treated in the intensive care unit did not survive the infection during this period. Informed consent from relatives for postmortem investigation was denied in 1/6 of the cases, resulting in an acceptance rate of 84%. The baseline characteristics of the patients are summarized in Table 1. The procedure was performed four times in a single-bed room (cases #1–4) and once in a multibed room (case #5). The time intervals of the action steps from the patient’s death until the end of sampling in the intensive care unit are shown in Figure 3. The mean time from patient death to relatives’ consent was 32 min (24–39), and that from consent to informing the TPT was 8 min (4–15). The time interval between the arrival of the TPT and the ICU was 162 min (129–210). The ultrasound examination required a mean of 13 min (5–16), and the following biopsy procedure required a mean of 54 min (44–75). The mean PMI was 215 min (case #1: 173, #2: 205, #3: 184, #4: 261, and #5: 254). In total, 318 samples were obtained (62 samples per deceased, on average), resulting in an average expenditure time of 50 s per biopsy.

### 3.1. Quality Assessment

A total of 112 samples were formalin-fixed, paraffin-embedded, and assessed for quality according to the quality criteria described above. The overall representativity for standard organ samples was 96.4% (106/110), and the mean biopsy length was 11 mm (4–25). Organ-specific results for representativity and biopsy length are presented in Table 2. One puncture of the left lung did not retrieve the lung parenchyma but retrieved the myocardial tissue (case #3). Tissue from the abdominal aorta and pancreas could not be successfully retrieved in case #2, most likely due to the high abdominal circumference (BMI 29) and insufficient length of the biopsy needle. No bone marrow was retrieved in case #3. Six biopsies were punctured outside the standard protocol due to special clinical questions or conspicuous imaging findings: tibial vein (suspected leg vein thrombosis), coronary artery (coronary artery sclerosis), pulmonary artery (suspected pulmonary artery embolism), arteria coronaria sinistra (coronary artery sclerosis), right atrium (thrombus), and abdominal lymph nodes (lymphoma). One pancreatic puncture retrieved not only pancreatic tissue but also gastric mucosa from the puncture channel. Organ-specific material was obtained from all the punctures except the lymph nodes. All biopsies contained well-preserved tissue without significant signs of autolysis under light microscopy (Figure 4B). Tissue vitality was particularly evident in the pancreas and gastric mucosa, which are prone to rapid postmortem autodigestion. Nevertheless, some samples from the kidneys, liver, and lungs contained necrotic cells (tubular epithelium, hepatocytes, and pneumocytes, respectively), which could be attributed to shock-related (kidney and liver) or disease-related (lung and liver) cell death.

At the ultrastructural level, EM revealed signs of incipient autolysis, such as disintegration of cellular membranes (Figure 4B and Figure 5). However, their degree was significantly lower than that of the materials derived from conventional autopsies. For example, in kidney samples, glomerular morphology was largely unaffected with partially discernible podocyte foot processes, basement membranes, and endothelial fenestrae (Figure 4B pictures).

### 3.2. Imaging and Histopathological Findings

The postmortem ultrasound and histopathological findings are summarized in Table 3. Typical ultrasound findings of COVID-19 (peripheral consolidation and B-lines) were observed in all the cases. The histomorphology of the lung samples (*N* = 37) revealed various stages of diffuse alveolar damage, microthrombi, metaplasia, and bacterial superinfection. Signs of shock-related multi-organ failure of the liver and kidneys (general hyperemia, necrotizing liver congestion, cholestasis, bile cast nephropathy, and acute tubular injury) were observed in all cases. Alongside the typical COVID-19-associated changes, we detected a fungal infection in case #2, causing renal and splenic infarctions with microscopic signs of *Aspergillus* spp. sepsis. Apart from the signs of acute disease, we also observed additional organ changes, such as lymphocyte depletion of the white pulp in the spleen in all cases (5/5), inflammation of the conjunctiva and skin, and parotid lipomatosis (3/5). Exemplary ultrasound pathologies and the corresponding histopathological findings are shown in Figure 6. In case #3, a high viral load was observed at the time of death. To study the distribution of SARS-CoV-2, samples from the lungs, pleura, liver, kidneys, parotid gland, conjunctiva, and skin were analyzed independently of microscopic pathologies. SARS-CoV-2 could be detected in different organs and cell types by EM and RNA in situ hybridisation, particularly in the lungs (pneumocyte type 1, alveolar macrophages, endothelium, and mesothelium) as well as in unexpected organs/cells such as acinar cells of the parotid gland and pancreas, kidneys (endothelium and tubular epithelium), and conjunctival epithelium. Exemplary EM images are illustrated in Figure 5.

### 3.3. Safety (Hygiene Control)

During the hospital stay, the patients were considered infectious, and SARS-CoV-2 was detected by repeated swabs. The last positive swabs (tracheal fluid) documented ante mortem (a.m.) were as follows: case #1 < 500 Geq/mL (6 days a.m.); case #2, 1950 Geq/mL (day of death); case #3, 700 Geq/mL (6 days a.m.); case #4, 18,347 Geq/mL (12 days a.m.); and case #5 < 500 Geq/mL (41 days a.m.). After death, SARS-CoV-2 was detected in (deep) nasopharyngeal swabs: case #1, 4786 GeQ/mL; case #2, 5491 Geq/mL; case #3, 16.613 Geq/mL; and case #4, 885 Geq/mL. Case #5 was negative (ORF/CT > 40). To evaluate the hygienic safety of the postmortem biopsy procedure, 60 swabs were collected after the procedure. No macroscopically visible contamination on the surface was visible to the naked eye. Among the 55 localizations, 54 were negative for SARS-CoV-2 with an ORF/CT > 40. In case #4, viral RNA was detected at a very low level (84 Geq/mL, CT/ORF 37.22) from the ultrasound transducer.

## 4. Discussion

Data on ultrasound-guided minimally invasive autopsy (US-MIA) in deceased critically ill COVID-19 patients are scarce. In our proof-of-concept study using MIA directly in the ICU, we demonstrated the feasibility of obtaining non-autolyzed material bedside from deceased critically ill patients for further advanced investigation.

Despite the benefits of clinical autopsies in diagnostics and research, not only in novel ways but also in describing new aspects of other infectious diseases such as influenza or yellow fever, a continuous decline in clinical autopsies over the last decades has been observed [13,20]. This development is partly due to new advancements in imaging technologies, easy-to-process biomarkers, and genetic material. This contrasts with the need for any novel disease to obtain information on organ involvement and pathogenesis to develop adequate therapies. To comply with these requirements, high-throughput multiorgan analyses of proteomics, genomics, and microbiomics are the most appropriate tools [21]. This need was apparent during the COVID-19 pandemic for depicting SARS-CoV-2 interactions between host organs and tissues. Conventional autopsy has the potential to provide a large-scale analytical approach to uncovering the pathogenesis of many acute and chronic diseases. Owing to the lack of conventional autopsy studies over the last few decades, studies on the mechanisms underlying COVID-19 pathogenesis are scarce [22]. In a recent plea, Layne et al. remind us of the dire need of the medical research community to prioritize autopsies for direct tissue studies that will help “to understand how SARS-CoV-2 causes severe disease” [23].

Traditional, century-old laborious autopsy practices represent a major disincentive; lack of standard operating procedures (SOPs), workforce cuts (pathologists and technical personnel), economic restrictions, strengthened safety regulations for infectious diseases, and concerns of patients/relatives regarding autopsies may explain the decline of conventional autopsies. Additionally, long postmortem intervals (PMI—the time between death and autopsy) are often more than 24–48 h, resulting in advanced autolysis and reduced usability for techniques such as electron microscopy, molecular assays (e.g., genomics, proteomics, and microbiomics), and cell culture models (e.g., spheroids and organoids) [21].

This contrasts with autopsy initiatives, coined as rapid research autopsy programs (RAP), which focus on beneficial, structured, and strengthened autopsy procedures [24]. Although RAP is labor-intensive and difficult to implement owing to logistics challenges, it has paved the way for another new autopsy approach. Bedside MIA represents a strategy for postmortem tissue procurement that complies with the above-mentioned requirements and offers hitherto unrecognized potential for scientific investigation. The US-MIA approach replaces the laborious opening and closing of major body cavities and organ preparation [25,26,27].

Another benefit is that ultrasound examinations and expertise are part of the daily routine in several clinical disciplines, including ultrasound-guided biopsies of the liver and kidneys. Thus, in combination with the broad availability of ultrasound machines, there is potential for a new way to obtain specific postmortem materials in specific contexts, such as cancer research. Notably, the procedure is faster than a conventional autopsy, and an increased frequency of repeated handling operations over time may enable quick and appropriate manual and technical procedures. Therefore, the MIA procedure saves time and workforce while preventing the spread of microorganisms during the procedure.

Additionally, high acceptance rates (84% in our study) for MIA stand out among relatives compared with conventional autopsies [28]. US-MIAs were first performed in countries with limited access to “health resources” in Africa and South America [25]. The potential of MIAs has been proven by studies on infectivity, diagnostic validity, and resource savings [25,28]. During the COVID-19 pandemic, MIA was also practiced in other countries, often with the adjunction of ultrasound to guide tissue sampling [29,30]. Histopathology of biopsies retrieved by MIA showed tantamount diagnoses compared to the results from conventional autopsies. Additionally, postmortem core biopsies have excelled in tissue preservation, which is advantageous for COVID-19 research. However, most MIA approaches do not follow structured sampling or tissue-handling protocols. In this study, we aimed to develop an interdisciplinary MIA approach to refine targeted tissue sampling using ultrasound guidance combined with a standardized tissue processing protocol. To minimize the PMI, US-MIA was conducted directly at the bedside of the ICU for critically ill COVID-19 patients who died.

The evaluation of this novel approach includes general feasibility, efficiency, safety, and sample quality. The feasibility analysis showed that the study protocol fit well with the clinical conditions. The acceptance of US-MIA by relatives exceeded our expectations, probably because of the less invasive nature of the procedure and the cogency of the doctors in charge who were acquainted with the details of the procedure. The efficiency of the method was broken down to the evaluation of tissue quality, which, owing to short PMIs, showed optimally preserved morphology at the cellular and subcellular levels, allowing viral detection using EM (SARS-CoV-2) and the RNAScope assay [31].

The safety of the procedure for viral contamination was tested by using multiple swabs at the end of each US-MIA. Except for one positive swab from the ultrasound transducer that had been in direct contact with the deceased patient’s puncture sites, all other swabs were virus-negative, which is in accordance with a previous study on the risk of SARS-CoV-2 contamination during autopsies of COVID-19 patients [32]. The diagnostic output of our bedside US-MIA revealed COVID-19-specific and non-specific organ pathologies similar to those detected by conventional autopsies. The postmortem US findings were comparable to those detected in patients with COVID-19. In some cases, postmortem US has detected previously unknown pathologies. Finally, we detected SARS-CoV-2 in various unexpected organs and cell types of deceased COVID-19 patients, such as the parotid gland, conjunctival mucosa, and pancreas, raising the possibility of viral persistence, a finding similar to other RNA viruses such as influenza, where the virus persists in various organ niches.

## 5. Conclusions

Our proof-of-concept study underlines the feasibility of obtaining vital materials for further investigations using bedside US-MIA in deceased critically ill COVID-19 patients. The success of bedside US-MIA relies on interdisciplinary collaboration, rapid communication, and the use of clinical resources in a postmortem context.

## 6. Research in Context

### 6.1. Evidence before This Study

We searched PubMed for articles using the search terms “COVID-19”, “critically ill”, and “minimally invasive autopsy”. This search yielded case reports and case series describing minimally invasive autopsy (MIA) approaches for COVID-19 patients. However, a systematic proof-of-concept study in postmortem critically ill COVID-19 patients evaluating bedside MIA techniques in the nearest possible time frame after death in the ICU is missing.

### 6.2. Added Value of This Study

This proof-of-concept study used US-MIA performed on postmortem critically ill COVID-19 patients directly bedside in the ICU. We registered a high acceptance rate among the relatives. A short time frame between the patient’s death and the start of MIA is feasible, and autolytic-free tissues can be efficiently obtained from multiple organ sites, including atypical locations such as the parotid gland. These autolytic-free materials make it possible to use deep analysis diagnostics, such as electronic microscopy and RNA-expressing techniques, for the detection of SARS-CoV-2 in unexpected organs.

### 6.3. Implications of All the Available Evidence

Minimally invasive approaches can help to maintain declining autopsy rates. Moreover, bedside tissue sampling is an efficient and safe method for obtaining high-quality vital tissues for diagnostic and research purposes, not only for future pandemics but also for cancer research and new technologies, such as organoids.

## Figures and Tables

**Figure 1 diagnostics-14-00294-f001:**
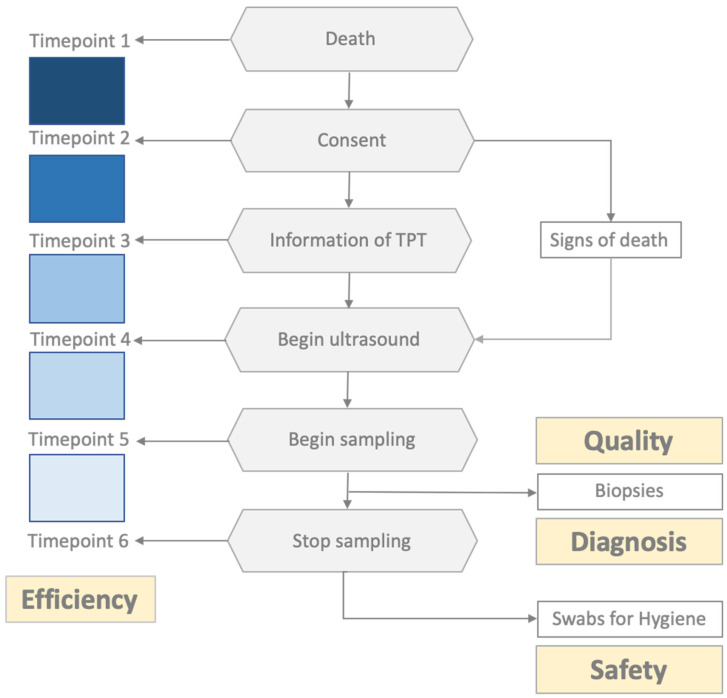
Study design. Investigational steps are marked as gray hexagons; key study objective measurement parameters are marked as yellow boxes.

**Figure 2 diagnostics-14-00294-f002:**
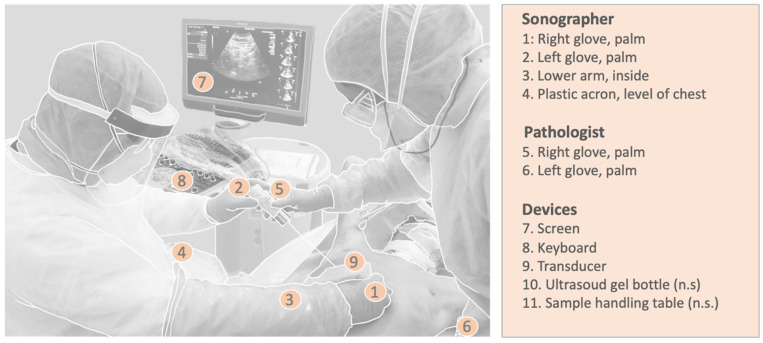
Swab collection localizations (n.s. = not shown).

**Figure 3 diagnostics-14-00294-f003:**
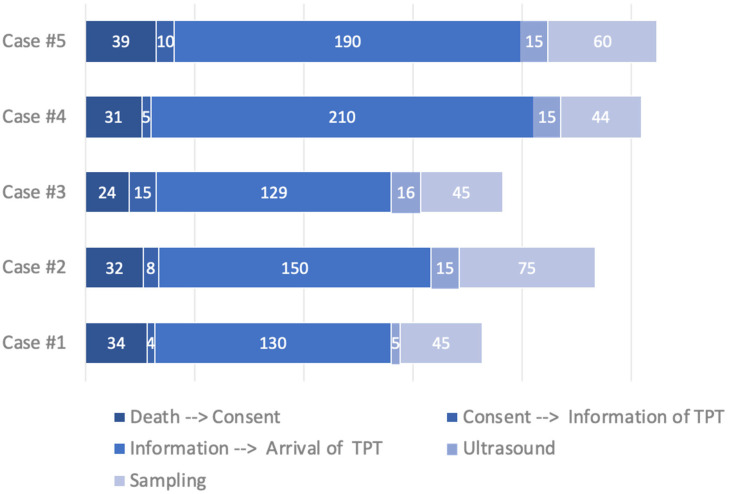
Time intervals (min) of the action steps from the patient’s death until the end of sampling at the intensive care unit.

**Figure 4 diagnostics-14-00294-f004:**
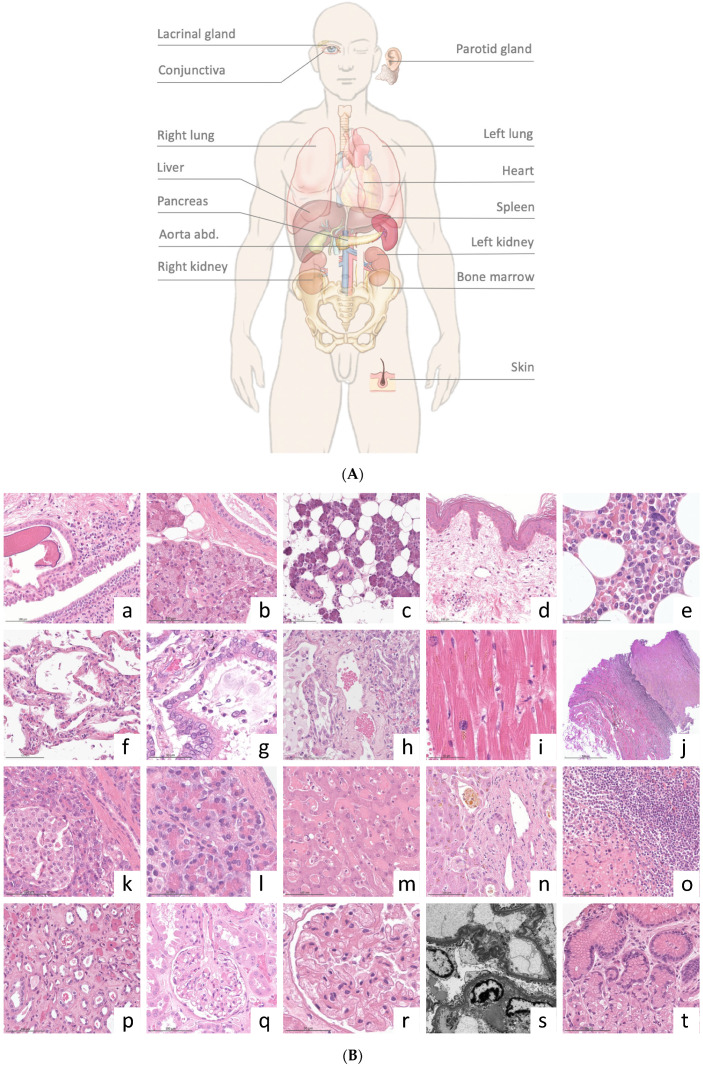
(**A**) Multiorgan sampling locations. (**B**) Exemplary microscopic morphology: (**a**) conjunctiva, (**b**) lacrimal gland, (**c**) parotid gland, (**d**) skin, (**e**) bone marrow, (**f**) alveolar lung, (**g**) alveolar lung with ciliated bronchiolization, (**h**) intrapulmonal vessel with inflammation, (**i**) myocardial lipofuscinosis, (**j**) abdominal aorta with intima fibrosis, (**k**,**l**) exocrine, endocrine, and ductal pancreatic epithelium, (**m**,**n**) liver with acute congestion (**m**) and cholestasis (**n**), (**o**) splenic white pulp, (**p**) renal tubular injury and microthrombi, (**q**,**r**) renal glomerulus, (**s**) renal glomerulus (electron microscopy), and (**t**) gastric mucosa. Stains: H&E (**a**–**i**,**k**–**q**,**s**) and EvG (**j**).

**Figure 5 diagnostics-14-00294-f005:**
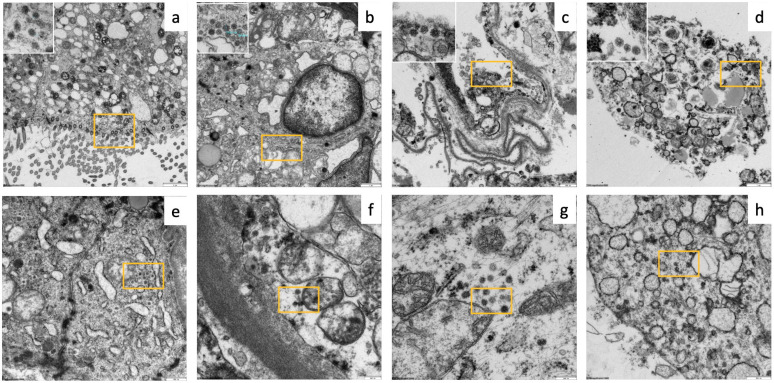
Electron microscopy detecting SARS-CoV-2 virions (yellow squares/insets) in case #3: (**a**) ciliated epithelium of the olfactory mucosa (mag. 3000×), (**b**) glandular cell of the olfactory mucosa (mag. 5000×), (**c**) pneumocyte type I (mag. 8000×), (**d**) alveolar macrophages (mag. 6000×), (**e**) acinar cell in a parotid gland (mag. 10,000×), (**f**) endothelial cell of a kidney (mag. 20,000×), (**g**) conjunctival epithelium (mag. 25,000×), and (**h**) pancreatic acinar cell (mag. 12,000×). Blue line indicates diameter size of about 70 nm.

**Figure 6 diagnostics-14-00294-f006:**
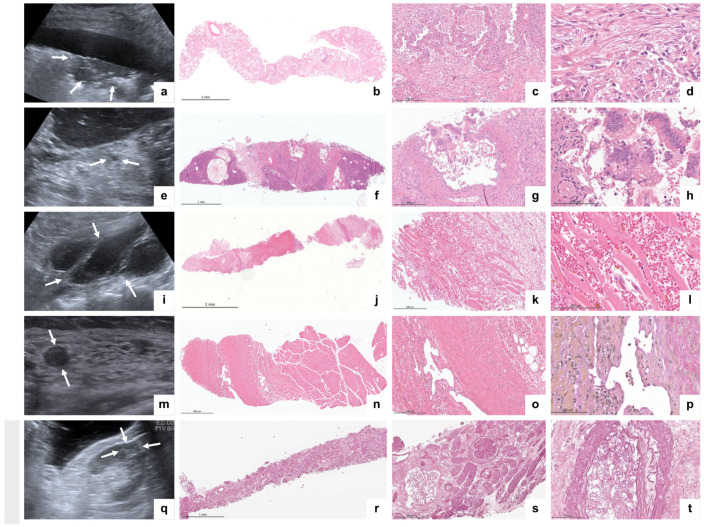
Exemplary ultrasound images and corresponding histopathological findings. (**a**) Peripheral lung consolidation (arrows) and pleural effusion, (**b**–**d**) lung biopsy with proliferative DAD (H&E), (**e**) pancreatic cyst (arrows), (**f**–**h**) pancreatic intraepithelial neoplasia 1a (H&E), (**i**) retroperitoneal hematoma (arrows), (**j**–**l**) skeletal muscle with hemorrhage (H&E), (**m**) vein thrombosis (arrows), (**n**–**p**) intramuscular vein thrombosis with organization ((**n**,**o**), H&E and (**p**), EvG)), (**q**) kidney infarction (arrows), and (**r**–**t**) a kidney with intravascular fungi (H&E).

**Table 1 diagnostics-14-00294-t001:** Patient characteristics and clinical findings.

	Case #1	Case #2	Case #3	Case #4	Case #5
Age (years), sex	75, male	65, male	78, male	73, female	87, male
BMI	27.1	29	25.5	24.8	24.5
ICU stay (days)	24	2	26	9	32
Hospital stay (days)	24	43	26	17	34
SOFA score	12	8	13	8	16
Mechanical ventilation (days)	24	2	26	9	32
Vaccination status until time of death	2×, BioNTech/Pfizer	2×, BioNTech/Pfizer	2×, BioNTech/Pfizer	2×, BioNTech/Pfizer	3×, BioNTech/Pfizer
COVID-19-specific medication	Dexamethasone and casirivimab/Imdevimab	Dexamethasone and casirivimab/Imdevimab	Dexamethasone and casirivimab/Imdevimab	Dexamethasone and casirivimab/Imdevimab	Dexamethasone
Medical history	AH, T2DM, and FSGS (under therapy with ciclosorin and rituximab)	AH, CHD, and diverticuosis history of DLBCL (RD negative) and tMDS/AML (under therapy, RD negative)	T2DM, CHD, atherosclerosis, and history of prostate cancer (RD negative)	AH, CHD, idiopathic lung fibrosis, cardiac enlargement, and idiopathic MDS (high risk)	AH, COPD, thrombosis of left saphenous vein and intramuscular veins, and pulmonary artery embolism
Imaging findings (CT)	CO-RADS 6	CO-RADS 6 (left lung); renal and splenic infarctions	CO-RADS 6	CO-RADS 6	Suspicious for COVID-19 (CO-RADS 4)
Clinical diagnosis leading to death	COVID-19 pneumonia, ARDS, and hypoxaemia	COVID-19 pneumonia and suspected fungal infection	COVID-19 pneumonia, ARDS, respiratory global insufficiency, suspected HSV, and fungal infection	COVID-19 pneumonia and septic multiorgan failure	COVID-19 pneumonia, ARDS, suspected bacterial superinfection, and hypoxaemia

ARDS, acute respiratory distress syndrome; AH, arterial hypertension; CHD, coronary heart disease; COPD, chronic obstructive pulmonary disease; CO-RADS 6, typical COVID-19 findings on CT in association with positive SARS-CoV-2; CO-RADS 4, suspicious but not extremely typical COVID-19 findings on CT; DLBCL, diffuse large B-cell lymphoma; and FSGS, focal segmental glomerulosclerosis. ICU = intensive care unit; SOFA = sequential organ failure assesment; T2DM = Type 2 diabetes mellitus; tMDS = therapy-related myleodysplastic syndrome; RD = residual disease; AML = Acute myeloid leucemia; CT = computed tomography.

**Table 2 diagnostics-14-00294-t002:** Organ-specific results on representativity and biopsy lengths.

Organ/Target	Representativity (%)	Mean Length of Biopsy (mm)
Right lung	21/21 (100)	14 (8–20)
Left lung	15/16 (94)	10 (4–15)
Liver	15/15 (100)	14 (11–17)
Right kidney	8/8 (100)	11 (5–15)
Left kidney	5/5 (100)	14 (3–15)
Heart	9/9 (100)	13 (5–25)
Pancreas	5/6 (83)	10 (8–12)
Abdominal aorta	4/5 (80)	8 (4–10)
Spleen	5/5 (100)	14 (9–17)
Lacrimal gland/conjunctiva	5/5 (100)	8.8 (6–11)
Parotid gland	5/5 (80)	9.6 (7–12)
Skin	5/5 (100)	10 (8–14)
Bone marrow	4/5 (80)	12 (9–16)
Total	106/110 (96)	11 (4–25)

**Table 3 diagnostics-14-00294-t003:** Postmortal ultrasound and histomorphological findings.

	Ultrasound	Pulmonary Histomorphology	Extrapulmonary Histomorphology
Case #1	Bilateral pleural effusions. Bilateral multiple peripheral lung consolidation. Hepatomegaly. Hepatic steatosis grade 1. Hydrops of the gallbladder.	Extensive DAD (mainly proliferative; partly exudative) and microthrombi.	Subacute and acute hepatic congestion; non-alcoholic fatty liver disease (5%). Interstitial pancreatic fibrosis; pancreatic acinar atrophy. Pancreatic intraepithelial neoplasia (low grade). Splenic red pulp expansion; white pulp hypoplasia. Acute renal tubular injury, microthrombi, and minimal focal segmental glomerulosclerosis. Myocardial interstitial fibrosis. Chronic conjunctivitis. Parotid lipomatosis. Dermal perivascular lymphocytic inflammation.
Case #2	Pleural effusion (right); small lung consolidations (left). Ascites. Hepatomegaly and aerobilia. Liver cyst (9 mm). Bilateral kidney and spleen infarctions.	Beginning DAD (exudative). Microthrombi. Mild interstitial fibrosis. Anthracosis. Emphysema.	Fungal sepsis (*Aspergillus* spp.) with the colonization of renal infarctions. Hepatocellular and canalicular cholestasis, chronic portal inflammation, hemosiderosis, bile cast nephropathy, acute renal tubular injury, splenic hemosiderosis, white pulp hypoplasia, myocardial interstitial fibrosis (infarct-like) bone marrow hypoplasia, edema, and hemosiderosis.
Case #3	Pleural effusion (right); bilateral multiple peripheral lung consolidation Aerobilia. Pancreatic lipomatosis. Aortic sclerosis. Retroperitoneal hematoma.	Extensive DAD (exudative and proliferative). Interstitial fibrosis. Bacterial superinfection. Microthrombi. No HSV or fungi.	Hepatocellular single cell necrosis, cholestasis, sinusoidal neutrophils, and non-alcoholic fatty liver disease (5%). Splenic red pulp expansion; white pulp hypoplasia. Acute renal tubular injury. Renal interstitial fibrosis; focal global glomerulosclerosis. Pancreatic lipomatosis. Retroperitoneal hematoma. Parotid lipomatosis. Lymphocytic inflammation of conjunctiva and lacrimal gland dermal perivascular lymphocytic inflammation.
Case #4	Bilateral pleural effusions. Bilateral multiple peripheral lung consolidation. Ascites. Hepatomegaly; aerobilia; splenomegaly. Pancreatic lipomatosis. Pancreatic cyst (5 mm).	Extensive DAD (exudative and proliferative); microthrombi pulmonary congestion.	Acute hepatic congestion; sinusoidal neutrophils. Non-alcoholic fatty liver disease (5%). Pancreatic intraepithelial neoplasia (low grade). Splenic red pulp expansion; white pulp hypoplasia. Acute renal tubular injury. Myocardial hypertrophy. Myelodysplastic syndrome with blast excess 2 (18% blasts). Chronic conjunctivitis. Dermal perivascular lymphocytic inflammation. Parotid lipomatosis.
Case #5	Bilateral multiple peripheral lung consolidation. Hepatic steatosis grade 1. Aortic sclerosis. Muscle vein thrombosis (*M. soleus*).	DAD (exudative and proliferative); microthrombi; infarct-like hemorrhage; bacterial superinfection	Severe acute hepatic congestion, sinusoidal neutrophils, and hepatocellular cholestasis. Non-alcoholic fatty liver disease (2%). Acute renal tubular injury, focal global glomerulosclerosis, and renal microthrombi. Pancreatic lipomatosis. Splenic red pulp expansion; white pulp hypoplasia. Chronic and acute conjunctivitis. Dermal perivascular lymphocytic inflammation. Aortic sclerosis. Muscle vein thrombosis in organization.

DAD = diffuse alveolar damage, HSV = Herpes simplex virus, and spp. = species.

## Data Availability

The data presented in this study are available on request from the corresponding author. The data are not publicly available due to local legal restrictions.

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
