# Peer review of "Postmortem Minimally Invasive Autopsy in Critically Ill COVID-19 Patients at the Bedside: A Proof-of-Concept Study at the ICU"

_diagnostics, 2024, doi:10.3390/diagnostics14030294_

Round 1

Reviewer 1 Report

Comments and Suggestions for Authors

The paper Postmortem minimally invasive autopsy in critically ill COVID-19 patients at the bedside: a proof-of-concept study at the ICU, signed by a group of 20 authors, present pretty good work. The language used by authors is clear, concise and explicite.

However some points must be clarified and completed.

Thus,

1.Lines 94-95: The Ethics mention: As from the approval of the Ethics Board the date is missing (Ref. 225/20S)  I suggest you to mention the date in order to see the chronology of study stages. It would be very useful also if you mention the German legal framework on dealing with dead bodies and if you had some special provisions or derogations during the pandemic in this field

2. Line 90: When you mention about the next of kin's consent, you refer to oral consent or some form of written consent? Please explain what type of consent you obtained and how you have obtained it.

3. Figure 4 should be reframed, if possible split it in 2 different figures and complete all pictures with corresponding notes: a-s. To be clear, on every small picture you must insert the corresponding letter. Like now these pictures are very small and the the scale bar is completely missing. Please correct and complete this missing information, otherwise the information contained in the pictures is not accurate.

4. Figure 5: Please insert the corresponding letter on each image and complete the description accordingly. 

5. Figure 6: The scale bar is missing and the image cannot be read correctly. Please complete it.

6. As the text is full of abbreviations, I suggest authors to introduce a table of abbreviations before References section, in order to facilitate the lecture of the paper.

7. I also suggest authors to improve the list of references with some recent publications in this field of research, for example: https://doi.org/10.3390/pathogens10040412; https://doi.org/10.3390/jpm13020279; etc

Comments on the Quality of English Language

I have no observations

Author Response

Dear editorial team, dear reviewers,

Thank you for reviewing our paper and the chance to send you a revision of our manuscript.

We would like to thank you for the helpful comments to improve the quality of our work!

We performed a point by point revision and highlighted the changes in the manuscript.

Reviewer 1

The paper Postmortem minimally invasive autopsy in critically ill COVID-19 patients at the bedside: a proof-of-concept study at the ICU, signed by a group of 20 authors, present pretty good work. The language used by authors is clear, concise and explicite.

However some points must be clarified and completed.

Dear reviewer #1, thank you for your helpful comments!

1.Lines 94-95: The Ethics mention: As from the approval of the Ethics Board the date is missing (Ref. 225/20S, date: 13.12.2021)  I suggest you to mention the date in order to see the chronology of study stages. It would be very useful also if you mention the German legal framework on dealing with dead bodies and if you had some special provisions or derogations during the pandemic in this field. 

Thank you, we agree, this is a relevant point. We included this in the methods section.

The basis for conducting a clinical autopsy is the consent of the relatives, whereby there are neither nationwide nor statewide uniform regulations regarding the specific procedure. The consent process at the MRI involves informing and obtaining consent from the nearest relative, and must be documented in writing by the relative's signature on the corresponding form. Deceased individuals without relatives or deceased individuals with legal guardians without a guardianship directive beyond death cannot, in principle, undergo autopsy according to the specific local regulations of the MRI.

The consent of the next of kin for an autopsy of a deceased person due to COVID-19 may be given orally (e.g., over the phone) and does not require a personal signature. The written documentation of consent may be carried out by the informing physician, providing the name of the relative, contact information, and the date and time of the consent.

  1. Line 90: When you mention about the next of kin's consent, you refer to oral consent or some form of written consent? Please explain what type of consent you obtained and how you have obtained it.

Thank you, please see also point 1.

The consents of the next of kin for the minimally invasive autopsies of the deceased persons in this study was given orally and documented by the informing physician (see above).

  1. Figure 4 should be reframed, if possible split it in 2 different figures and complete all pictures with corresponding notes: a-s.

We agree and rearranged the figures.

Figure was split into 2, Figure 4a: Multiorgan sampling locations , Fig 4b, Exemplary microscopic morphology: a, conjunctiva, b, lacrimal gland, c, parotid gland, d, skin, e, bone marrow, f, alveolar lung, g, alveolar lung with ciliated bronchiolization, h, intrapulmonal vessel with inflammation, i, myocardial lipofuscinosis , j, aorta abdominalis with intima fibrosis, k/l, exocrine, endocrine and ductal pancreatic epithelium, m/n, liver with acute congestion (m) and cholestasis (n) , o, splenic white pulp, p, renal tubular injury and microthrombi, q/r renal glomerulus, s, renal glomerulus (electron microscopy), t, gastric mucosa. Stainings: H&E (a-I, k-q, s), EvG (j).

To be clear, on every small picture you must insert the corresponding letter. Like now these pictures are very small and the the scale bar is completely missing. Please correct and complete this missing information, otherwise the information contained in the pictures is not accurate.

Letters were added. The scale bar has been added to each photo. The scale bars are quite small due to the small image format, but they can be enlarged as desired after downloading a PDF.

  1. Figure 5: Please insert the corresponding letter on each image and complete the description accordingly. 

Letters were added. The scale bar has been added to each photo

  1. Figure 6: The scale bar is missing and the image cannot be read correctly. Please complete it.

The scale bar and magnification has been added to each image. The number images have been reduced to enable a more detailed, larger display. Scales and magnification have been added for each photo.

  1. As the text is full of abbreviations, I suggest authors to introduce a table of abbreviations before References section, in order to facilitate the lecture of the paper.

This is  a good point. We added this before the references.

  1. I also suggest authors to improve the list of references with some recent publications in this field of research, for example: https://doi.org/10.3390/pathogens10040412; https://doi.org/10.3390/jpm13020279; etc

We included some more references as suggested

Once again, thank you for the possibility to revise our manuscript. We hope you are satisfied with the changes we made. We are looking forward for publication of our manuscript in Diagnostics.

With kind regards

Tobias Lahmer

Reviewer 2 Report

Comments and Suggestions for Authors

Minimally invasive autopsy approaches are becoming a new gold standard, which is increasingly evident in the COVID era. ICU is a part of a healthcare system with maximal scrutiny and the most ancillary approach. Usually, antemortem diagnostic certainty about ICU patients is correlated with the length of ICU stay, which was not the case in this study (it seems that patients were included in the minimally invasive autopsy cohort irrespective of the length of stay, and we could benefit of some  correlation  regarding the correlation of the LoS from Table 1-  with findings in "minimally invasive PM."-

1.      Ln's 32/33 – define "length": and again in the Table 2 – be coherent – what is the length of biopsy?

2.      Please, adjust the width of column #1 in Table 1 – so that it is content-adjusted

3.      When you say “vaccination status” – you have to ne more specific

4.      All the abbreviations should be given in full eirst time they are mentioned

Author Response

Dear editorial team, dear reviewers,

Thank you for reviewing our paper and the chance to send you a revision of our manuscript.

We would like to thank you for the helpful comments to improve the quality of our work!

We performed a point by point revision and highlighted the changes in the manuscript.

Reviewer 2

Dear reviewer #2, thank you for your helpful comments!

Minimally invasive autopsy approaches are becoming a new gold standard, which is increasingly evident in the COVID era. ICU is a part of a healthcare system with maximal scrutiny and the most ancillary approach. Usually, antemortem diagnostic certainty about ICU patients is correlated with the length of ICU stay, which was not the case in this study (it seems that patients were included in the minimally invasive autopsy cohort irrespective of the length of stay, and we could benefit of some  correlation  regarding the correlation of the LoS from Table 1-  with findings in "minimally invasive PM."-

  1. Ln's 32/33 – define "length": and again in the Table 2 – be coherent – what is the length of biopsy?

This is correct. We included the length of biopsy (mm)

  1. Please, adjust the width of column #1 in Table 1 – so that it is content-adjusted

            We´ve adjusted the table following your suggestion.

  1. When you say “vaccination status” – you have to ne more specific

As you can see we included already in the table how often and the kind of vaccination. We more specified the vaccination status including until time of death. Hope you find this helpful.

  1. All the abbreviations should be given in full first time they are mentioned

Is now included , see also above reviewer #1

Once again, thank you for the possibility to revise our manuscript. We hope you are satisfied with the changes we made. We are looking forward for publication of our manuscript in Diagnostics.

With kind regards

Tobias Lahmer

Round 2

Reviewer 1 Report

Comments and Suggestions for Authors

Dear Authors,

Thank you for taking into consideration my recommendations and suggestions! After you have have integrated reviewers comments and suggestions, the paper seems to be improved.

Anyway I have noticed some issues and I recommend to be taken into consideration.

Thus:

1. Line 118: 'After consent was obtained'..., I recommend 'After oral consent was obtained...' - for accuracy.

2. Line 200: 'world health organization' to be replaced with World Health Organization - this is the name of the institution. 

3. Lines 206 - 207: 'Patient enrollment for this study started on December 10, 2021, and ended on January 206 15, 2022'. I suggest this sentence to be moved within the Study design right after the first sentence there, before the inclusion criteria.. (line 89).

4. In the abbreviation list, I suggest to change  some abbreviated words with capitals. Eg. 'world health organization' - World Health  Organization

Author Response

Dear editorial team, dear reviewers,

Thank you for reviewing our paper and the chance to send you a revision of our manuscript.

We would like to thank you for the helpful comments to improve the quality of our work!

We performed a point by point revision and highlighted the changes in the manuscript.

Thank you for taking into consideration my recommendations and suggestions! After you have have integrated reviewers comments and suggestions, the paper seems to be improved.

Anyway I have noticed some issues and I recommend to be taken into consideration.

Thus:

  1. Line 118: 'After consent was obtained'..., I recommend 'After oral consent was obtained...' - for accuracy.

Is now included following your suggestion.

  1. Line 200: 'world health organization' to be replaced with World Health Organization - this is the name of the institution. 

Thank you for your advice and we have changed it following your suggestion.

  1. Lines 206 - 207: 'Patient enrollment for this study started on December 10, 2021, and ended on January 206 15, 2022'. I suggest this sentence to be moved within the Study design right after the first sentence there, before the inclusion criteria.. (line 89).

Thank you for this good advice, we´ve moved this sentence in the study design section, following your suggestion.

  1. In the abbreviation list, I suggest to change  some abbreviated words with capitals. Eg. 'world health organization' - World Health  Organization

We´ve changed some abbreviated words, following your suggestion

Once again thank you for the possibility to revise our manuscript. We looking forward for publication in your journal.

With best regards

Tobias Lahmer